# ANLoC: An Anomaly-Aware Node Localization Algorithm for WSNs in Complex Environments

**DOI:** 10.3390/s19081912

**Published:** 2019-04-23

**Authors:** Pengfei Xu, Tianhao Cui, Lei Chen

**Affiliations:** 1School of Computer Science, Nanjing University of Posts and Telecommunications, Nanjing 210023, China; 1217043202@njupt.edu.cn (P.X.); 1018041227@njupt.edu.cn (T.C.); 2Jiangsu Key Laboratory of Big Data Security & Intelligent Processing, Nanjing University of Posts &Telecommunications, Nanjing 210023, China; 3College of Computer Science and Technology, Nanjing University of Aeronautics and Astronautics, Nanjing 210016, China

**Keywords:** wireless sensor networks, anomaly-aware node localization, low-rank matrix decomposition, mixture of Gaussians

## Abstract

Accurate and sufficient node location information is crucial for Wireless Sensor Networks (WSNs) applications. However, the existing range-based localization methods often suffer from incomplete and detorted range measurements. To address this issue, some methods based on low-rank matrix recovery have been proposed, which usually assume noises follow single Gaussian distribution or/and single Laplacian distribution, and thus cannot handle the case with wider noise distributions beyond Gaussian and Laplacian ones. In this paper, a novel Anomaly-aware Node Localization (ANLoC) method is proposed to simultaneously impute missing range measurements and detect node anomaly in complex environments. Specifically, by utilizing inherent low-rank property of Euclidean Distance Matrix (EDM), we formulate range measurements imputation problem as a Robust $\ell_{2,1}$-norm Regularized Matrix Decomposition (RRMD) model, where complex noise is fitted by Mixture of Gaussian (MoG) distribution, and node anomaly is sifted by $\ell_{2,1}$-norm regularization. Meanwhile, an efficient optimization algorithm is designed to solve proposed RRMD model based on Expectation Maximization (EM) method. Furthermore, with the imputed EDM, all unknown nodes can be easily positioned by using Multi-Dimensional Scaling (MDS) method. Finally, some experiments are designed to evaluate performance of the proposed method, and experimental results demonstrate that our method outperforms three state-of-the-art node localization methods.

## 1. Introduction

Wireless Sensor Networks (WSNs) have made great progress and have been widely used in various fields, such as environmental monitoring, intelligent transportation, target tracking, etc. [1,2,3]. The premise that these applications work well is accurate location information acquisition [4,5]. Up to now, many localization methods have been proposed, which could be divided into two categories [6]. One is called range-based localization method that could achieve more accurate positioning, but the computation and communication overhead is large while some hardware support is also required. The other is named range-free localization method that is generally suitable for low power and cost applications, yet their positioning accuracy is low.

In this paper, we mainly focus on the range-based localization methods, which could be described as follows: in WSNs applications, some sensor nodes are randomly disposed, and a few of them called anchors could get actual location information by GPS device or other equipment. Then, with the pair-wise range measurements between nodes and actual location of anchors, the location of all unknown nodes could be easily estimated. In general, range-based localization methods often depend on a large amount of accurate inter-node distance information. However, in practical situations, limited by the energy of sensors or the distribution of nodes in application scenario, only a small amount of inter-node distance information could be obtained [7]. Additionally, due to complex environments and uncertain hardware abnormality, the range measurements inevitably suffer from some errors, which lead to lower positioning accuracy. Usually, these errors can be regarded as a mixture of complex noise and anomaly. In reference [6], the complex noise is considered to be caused by environmental interference, malicious attacks, or other unpredictable factors, and is assumed to be a mixture of Gaussian noise with single distribution and outlier noise with single distribution. Correspondingly, Xiao et al. [6] proposed a method based on low-rank matrix completion to sift the complex noise by adopting both Frobenius-norm regularization and $\ell_{1}$-norm regularization. However, presetting the complex noise into two known noise types was too arbitrary. Actually, the noise distribution in practical applications was often unknown and showed a wider noise distribution beyond Gaussian and Laplacian ones. On the other hand, reference [8] handled one more general application scenario with the co-existence of complex noise and anomaly nodes, where anomaly nodes were defined as ones with abnormal transmission module or unpredictable hardware defects. In reference [8], a $\ell_{2,1}$-norm regularization term was employed to detect the anomaly from the corrupted range measurements. However, similar to reference [6], the so-called complex noise is still limited to be modeled as two known noise types, i.e., the mixture of outlier noise and Gaussian noise.

To address this limitation, we propose a novel Anomaly-aware Node Localization (ANLoC) method to simultaneously position the unknown nodes and probe the abnormal nodes in complex environments. Specifically, a Robust $\ell_{2,1}$-norm Regularized Matrix Decomposition (RRMD) model is constructed by introducing Mixture of Gaussian distribution and $\ell_{2,1}$-norm into the conventional low-rank Matrix Decomposition (MD) model, which can not only well fit the intrinsic low-rank property of Euclidean distance matrix, but also is robust against the node anomaly and a wider range of complex noise distributions beyond Gaussian and Laplacian noises. Our basic idea is to encode the node anomaly and the complex noise as a structural row/column sparsity matrix and a noise matrix, respectively, with entries that satisfy Mixture of Gaussian (MoG) distribution. Here, we prefer to employ MoG distribution as the general noise model due to its universal approximation property to any continuous distribution [9]. Such idea is inspired by some recent noise modeling works including Low-Rank Matrix Factorization (LRMF) [10] and Low-Rank Representation (LRR) [11], and has been verified to be effective in the complex noise scenarios. Note that although anomaly can also be seen as a continuous distribution that is approximated by MoG in theory, we still explicitly model it by using $\ell_{2,1}$-norm. The reason is to explicitly detect the location of anomaly node, and thus provide a basis for troubleshooting. Furthermore, based on the popular Expectation Maximization (EM) method [12], an efficient optimization algorithm for solving the proposed RRMD model is designed to obtain the true underlying Euclidean Distance Matrix (EDM). Finally, the actual coordinates of unknown nodes could be easily estimated by employing Multi-Dimensional Scaling (MDS) method [13], and the abnormal nodes can also be detected.

The primary contributions of our work can be summarized as follows.
A Robust $\ell_{2,1}$-norm Regularized Matrix Decomposition (RRMD) model is proposed to jointly estimate the missing range measurements and detect the node anomaly, which takes advantage of the potential relationship between two tasks which could help each other to achieve more accurate performance.The MoG distribution is employed to fit the unknown complex noise, which allows the proposed RRMD model to adaptively handle a wider range of noise beyond the existing methods. Meanwhile, an efficient optimization algorithm is designed to solve the proposed RRMD model by adopting the popular EM method.A novel Anomaly-aware Node Localization (ANLoC) method is proposed based on the RRMD model, and extensive experiments verify the superior positioning performance of the ANLoC method in the coexistence of node anomaly and complex noise.

The rest of this paper is organized as follows. In Section 2, we introduce the current research advances about the range-based node localization methods and low-rank matrix decomposition methods, respectively. Section 3 describes the notations used in this paper and some related mathematical foundations. In Section 4, the RRMD model is constructed and an optimization algorithm based on the EM method is designed to solve this model. This section also presents the Anomaly-aware Node Localization (ANLoC) method based on the proposed RRMD and the classic MDS. In Section 5, a series of simulation experiments are conducted to evaluate the performance of our proposed method. Finally, the conclusions are drawn in Section 6.

## 2. Related Work

### 2.1. Range-Based Node Localization

At present, the typical range-based localization methods include two steps: (1) using certain ranging methods to measure the distance between nodes, such as Time of Arrival (ToA), Time Difference of Arrival (TDoA) and Received Signal Strength Indicator (RSSI); (2) using the range measurements combined with the location of anchor nodes to calculate the position information of unknown nodes. The popular localization method called Maximum Likelihood (ML) is asymptotically efficient [14] with enough data records. Tomic et al. [15] built a new convex estimator that approximated the ML by applying efficient convex relaxations, which reduced the estimation errors. In references [16,17], WSNs localization problem is treated as a variant of EDM recovery problem or graph implementation problem. By using the range measurements between nodes and introducing the slack variable to convert non-convex quadratic distance constraints into linear constraints, the authors formulated the WSNs localization problem as a Semi-Definite Programming (SDP) problem, and designed an efficient optimization method to solve the proposed problem. References [18,19] employed the classical MDS method to map the distance relationship between wireless sensor nodes to low-dimensional space, and generated a relative coordinate map which fitted well the distance relationship between nodes. Then, a few anchor nodes were used to convert the relative position to the global position. However, the above methods did not work well in real application scenarios. In general, these methods required complete and accurate EDM. Unfortunately, due to the requirement of energy-saving and the effects of potential node anomaly and complex noise, the existing range-based node localization methods often suffer from incomplete and detorted range measurements. In addition, the ML method and SDP method could be only used to solve a small-scale problem because of its high computational complexity.

To address the above issues, some matrix completion based node localization methods have been proposed in recent years. Specifically, by taking advantage of the low-rank characteristics of EDM, Feng et al. [20] firstly formulated the range measurement imputation problem as a low-rank EDM Matrix Completion (MC) model, and designed an efficient optimization method to solve the proposed MC model. However, their work is limited to assume the errors contained in the range measurements are Gaussian noise, ignoring the existence of some complex errors caused by node hardware failure, multipath transmission, etc. In reference [6], another kind of errors called outliers was assumed to obey Laplacian distribution. Then, $\ell_{1}$-norm regularization was introduced to deal with it, which effectively improved the positioning accuracy. However, it was also too simple to preset the complex noise to these two known types, and the actual noises should be more complicated in practical applications. Recently, Liu et al. [21] proposed a Linear Bregman Iteration based matrix completion method to localize node position for WSNs. However, this method did not consider the actual scenario under the co-existence of complex noise and anomaly. More importantly, all these matrix completion based node localization methods inevitably involved Singular Value Decomposition (SVD) operation with heavy computation cost. To address these issues, an ANLoC method is proposed in this paper, which formulate the range measurements imputation problem as low-rank matrix decomposition instead of low-rank matrix completion model. Therefore, in the next section, we will introduce the related work on low-rank matrix decomposition.

### 2.2. Low-Rank Matrix Decomposition

Low-Rank Matrix Decomposition (LRMD) is an important technique in data science, which can uncover the latent manifold structures of data, and thus obtain a low dimensional compression representation. Recently, by decomposing the target matrix into the product of two low-rank matrices, LRMD has been widely used in various fields such as dimensionality reduction, clustering, and matrix recovery.

The original LRMD [22] model can be formulated as
(1)minU,V ||M−UVT||ℓp
where M∈ℝm×n is the target matrix to be approximated, U∈ℝm×r and V∈ℝn×r are two low-dimensional matrix variables (r<min(m,n)), ||·||ℓp denotes the ℓp-norm, and ℓ1-norm and ℓ2-norm are commonly used to make this model robust to Gaussian noise and outlier noise, respectively. When some elements of M are missing, the original LRMD could be changed into a matrix recovery model by adding an orthogonal projection operator [23], which could be formulated as:(2)minU,V ||PΩ(M−UVT)||ℓp
where PΩ(·) is an orthogonal projection operator defined as
(3)[PΩ(M)]ij={Mij,(i,j)∈Ω,0,otherwise,
where Ω⊆[m]×[n]([m]={1,2,…,m},[n]={1,2,…,n}) represents the index set of sampled elements. This model could be applied to various matrix recovery problems such as recommender system [24] and image representation [25], and has lower computational complexity than low-rank matrix completion model with nuclear norm minimization. However, ℓ1-norm and ℓ2-norm regularizations are only optimal when the noise follows a Gaussian or Laplacian distribution, which is not in line with the actual situation. To address this limitation, Meng et al. [10] proposed a noise-tolerant LRMD model based on Mixture of Gaussian distribution, which could be described as:(4)minU,V ||PΩ(M−UVT)||MoG
where ||·||MoG represents that each element of this matrix is modeled by Mixture of Gaussian distribution. Reference [10] has demonstrated the robustness of the MoG model to unknown noise. However, since this model does not explicitly consider the row/column-wise structural anomaly that the sampled matrix may suffer from, it cannot be directly applied to the node localization of WSNs in this paper.

## 3. Preliminaries

### 3.1. Notations

We first introduce the important notations used in this paper. All italic letters denote variables and non-italic letters denote constants. Bold uppercase letters denote matrices and bold lowercase letters denote vectors. Specifically, Xij denotes the scalar in the i-th row and j-th column of X. xi and xj represent the i-th row and j-th column of the matrix X respectively. Additionally, we denote the ℓp-norm, Frobenius-norm, and ℓ2,1-norm of matrix X∈ℝm×n as (∑i=1m∑j=1n|Xij|p)1/p,||X||F= (∑i=1m∑j=1nXij2)1/2, and ||X||2,1= ∑i=1m(∑j=1nXij2)1/2, respectively. XT denotes the transpose of matrix  X. N(0,σ2) denotes a Gaussian distribution with mean 0 and variance σ2. Finally, ⊙ represents the Hadamard product of two matrices.

### 3.2. Mathematical Foundation

**Definition** **1** [26]**:** (**Proximal Operator**) *Let*
F(X)
*be a real convex function defined on*
X∈ℝm×n*, for any*
τ, μ>0
*and constant matrix*
M∈ℝm×n*, the proximal operator could be defined as:*
(5)proxτF(X)(M)=argminXτF(X)+μ2||X−M||F2

**Definition** **2** [27]**: (Structural Thresholding Operator)**
*For any*
τ>0,M∈ℝm×n*, the proximal operator of*
ℓ2,1*-norm, i.e., structural thresholding operator, is defined as:*
(6)proxτ||X||2,1(M)=Jτ/μ(M)
*where*
(7)$${\left(J_{\tau /\mu }\left(M\right)\right)}^{i}=max\left\{||M^{i}||{}_{2}-\tau /\mu ,0\right\}\cdot M^{i}/||M^{i}||{}_{2},\;i=1,2,\ldots ,m$$

**Definition** **3** [28]**:** (**Lipschitz Continuous Gradient**) *A differentiable convex function*
F(X)
*defined on*
ℝm×n
*is said to have a Lipschitz continuous gradient*
ξ*, i.e., for*
∀X1,X2∈ℝm×n*, there exists a constant*
ξ>0*, such that*
(8)$${\left(J_{\tau /\mu }\left(M\right)\right)}^{i}=max\left\{||M^{i}||{}_{2}-\tau /\mu ,0\right\}\cdot M^{i}/||M^{i}||{}_{2},\;i=1,2,\ldots ,m$$

**Theorem** **1** [29]**:**
*Let*
$F_{1}$
*and*
$F_{2}$
*be two lower semi-continuous convex functions on*
X∈ℝm×n*, and*
F2
*is also a differentiable function with Lipschitz continuous gradient*
ξ*. Then, for a convex optimization problem defined as*
(9)$$\underset{X}{min}F_{1}\left(X\right)+F_{2}\left(X\right)$$
if $F_{1}+F_{2}$
*is mandatory and strictly convex, then for any initial value*
$X_{0}$ and 0<δ<1/ξ*, the iterative sequence*
$X_{k+1}$
*generated by the following Equation (10) is the unique solution of problem (9)*
(10)$$X_{k+1}=arg\underset{X}{min}\delta F_{1}\left(X\right)+\frac{1}{2}||X-{\left(X_{k}-\delta \nabla F_{2}\left(X_{k}\right))|\right|}_{F}^{2}$$

## 4. Anomaly-Aware Node Localization for WSNs

In this section, we first establish a RRMD model and employ EM method to optimize it, and then a novel ANLoC method is proposed based on the RRMD model.

### 4.1. Euclidean Distance Matrix Completion

#### 4.1.1. Problem Description and RRMD Model Construction

In a typical application scenario, n sensor nodes are randomly deployed in a specific area, whose coordinates can be formulated as X=[x1,x2,…,xn]∈ℝd×n (d usually be 2 or 3, indicating the dimension of the coordinate space). Then, the corresponding EDM matrix D∈ℝn×n could be calculated by
(11)Dij=||xi−xj||2=xiTxi+xjTxj−2xiTxj,i,j∈{1,2,…,n}
where Dij represents the squared Euclidean distance between the *i*-th sensor node and the *j*-th sensor node. Reference [8] has proofed that the rank of matrix D is at most d+2, which indicates the EDM matrix has an inherently strict low-rank property. Usually, in practical applications, only a few sensor nodes could obtain their accurate coordinates by loading GPS, and the position information of other unknown nodes need be indirectly calculated by employing some localization methods. Due to irregular node distribution and energy consumption limitation, only the partial range measurements could be collected. Based on the collected incomplete distance measurements, we can only establish a sampled EDM matrix with missing elements. Our goal is to estimate the coordinates of unknown nodes based on the sampled range measurements and the known coordinates of anchor nodes. Figure 1 illustrates the pipeline of node localization for WSNs in complex environments. As shown in Figure 1, the node localization process involves two main steps: EDM recovery and node positioning. At the first step of EDM recovery, by using our proposed RRMD algorithm, we can impute the missing range measurements and obtain the estimated complete EDM matrix, which is also called the true underlying EDM matrix, and then at the second step of node positioning, based on the estimated true underlying EDM matrix, we can further employ MDS method to position each unknown node. Compared to the second step of node positioning, the first step of EDM recovery is more critical and challenging. Specifically, in practical applications, limited by the complex environmental interference, malicious attacks, hardware malfunction, etc., the existence of errors in range measurements is unavoidable, which leads to the destruction of sampled EDM and reduction in localization accuracy. In this paper, we assume that the sampled EDM matrix M∈ℝn×n consists of the true underlying EDM component D^∈ℝn×n, the row structural anomaly matrix component R∈ℝn×n, the column structural anomaly matrix component C∈ℝn×n, and the noise matrix component N∈ℝn×n, which is illustrated in Figure 2. The noise is considered to be caused by environmental interference, malicious attacks, etc., which usually follows continuous complex distributions. Moreover, the existence of the anomaly is caused by abnormal transmission module or other unpredictable hardware defects. Specifically, when the receiving module of the nodes fails, it will cause row structural anomaly in the corresponding row in EDM, and when the sending module of nodes fails, the corresponding column in the EDM will have the column structural anomaly [8]. To address the above problems, we need to design an efficient method to obtain a true underlying EDM. Finally, based on this EDM, the actual coordinates of unknown nodes could be easily estimated by employing the classic MDS method.

Therefore, our primary goal is to design an effective EDM reconstruction method to impute the missing range measurements and de-noise the inaccurate ones. Intuitively, based on the low-rank characteristics of EDM, we can classify the EDM reconstruction problem into a standard matrix decomposition model, which is formulated as
(12)$$\underset{U,V}{min}\;||P_{\Omega }{\left(M-UV^{T})|\right|}_{F}.$$

However, this model can only be applied to the case of single Gaussian noise, and its reconstruction performance will decline dramatically when the sampled EDM is under co-existence of complex noise and anomaly. To this end, we employ MoG distribution and ℓ2,1-norm to improve the model, and then establish a Robust ℓ2,1-norm Regularized Matrix Decomposition (RRMD) model as follows:(13)minD^,R,C,N,U,V,Π,Σλ1||R||2,1+λ2||CT||2,1−logL(PΩ(N)|Π,Σ),s. t.  PΩ(M)=PΩ(D^+N+R+C),D^=UVT,∑k=1ncπk=1,πk≥0, k=1,2,⋯,nc,
where M∈ℝn×n denotes the sampled matrix, R∈ℝn×n and C∈ℝn×n represent row structural anomaly and column structural anomaly, respectively, U,V∈ℝn×(d+2) are two low-rank components of the true underlying EDM matrix D^, logL(PΩ(N)|Π,Σ) is the log-likelihood function, Π={π1,π2,…,πnc},Σ={σ12,σ22,…,σnc2} and nc are the parameters of MoG, and λ1 and λ2 are the tunable parameters. The key of this model is that we use MoG distribution to smooth any unknown noises and ℓ2,1-norm to detect node anomaly, respectively. The main motivations for this model are as follows. (1) First of all, in response to the complex noise, we introduce a Mixture of Gaussians distribution, which is treated as a universal approximator to any continuous density function. Therefore, the RRMD model is robust against a wider range of complex noise distributions beyond Gaussian and Laplacian noises. (2) Then, as the optimal measure of row sparsity matrix, ℓ2,1-norm could be used to smooth row-wise anomaly and the transpose of column-wise anomaly, thus the abnormal nodes can be detected. (3) Finally, in order to deal with the challenge of data missing, we choose the classic low-rank MD method, which can impute the missing elements with lower computational costs than the low-rank MC method.

In this paragraph, we introduce how to apply MoG distribution to the RRMD model. Without loss of generality, each element $N_{ij}\left(i,j=1,2,\ldots ,n\right)$ in the noise matrix N is considered to be from a Mixture of Gaussians distribution, which is defined as
(14)$$𝕡\left(N_{ij}\right)\sim \overset{n_{c}}{\underset{k=1}{\sum }}\pi_{k}N\left(N_{ij}|0,\sigma_{k}^{2}\right)$$
where $N\left(N_{ij}|0,\sigma_{k}^{2}\right)$ denotes the Gaussian distribution with mean 0 and variance $\sigma^{2}$. $n_{c}$ is the number of Gaussian components and $\pi_{k}\geq 0$ represents the mixing proportion where $\overset{n_{c}}{\underset{k=1}{\sum }}\pi_{k}=1$. Then, the likelihood function can be written as
(15)$$L\left(P_{\Omega }\left(N\right)|\Pi ,\Sigma \right)=\prod_{i,j\in \Omega }\sum_{k=1}^{n_{c}}\pi_{k}N\left(N_{ij},0,\sigma_{k}^{2}\right)$$
where $\Pi =\left\{\pi_{1},\pi_{2,\ldots ,}\pi_{n_{c}}\right\}$, $\Sigma =\left\{\sigma_{1}^{2},\sigma_{2}^{2},\ldots ,\sigma_{n_{c}}^{2}\right\}.$ Usually, we use the log-likelihood function instead of this likelihood function for convenient calculation. Then, our aim is to maximize the log-likelihood function to obtain the MoG parameters. Obviously, we can construct the final objective function expressed as Equation (13) by combining the negative log-likelihood function with the $\ell_{2,1}$-norm, and then minimize it to get all unknown variables in RRMD model.

#### 4.1.2. Optimizing RRMD via Expectation Maximization Method

In this section, we employ the popular EM method to solve the proposed RRMD model. Firstly, we explain the motivation for using this method. Obviously, in order to gain the estimated values of parameters Π,Σ, the log-likelihood function $logL\left(P_{\Omega }\left(N\right)|\Pi ,\Sigma \right)$ should be maximized. However, for many specific problems, parameters Π,Σ cannot be directly calculated due to complex expressions of log-likelihood functions. Fortunately, the EM method can be employed to solve such problems, which has been proven to be effective by many researches. Specifically, let $Z_{ijk}\in \left\{0,1\right\}$ be a set of hidden variables with $\sum_{k=1}^{n_{c}}Z_{ijk}=1\left(i,j=1,2,\ldots ,n\right)$, and $Z_{ijk}=1$ implies the noise $N_{ij}$ comes from the k-th Gaussian component. Therefore, the log-likelihood function can be re-written in a form that is easy to optimize as
(16)$$logL\left(P_{\Omega }\left(N\right)|\Pi ,\Sigma \right)\;=\sum_{i,j\in \Omega }\sum_{k=1}^{n_{c}}Z_{ijk}\left(log\pi_{k}-log\sqrt{2\pi }\sigma_{k}-\frac{1}{2\sigma_{k}^{2}}{\left(M_{ij}-u_{i}{}^{T}v_{j}-R_{ij}-C_{ij}\right)}^{2}\right)$$

Next, we will introduce the specific optimization process of estimating the parameters ($\hat{D},R,C,N,U,V,\Pi ,\Sigma$) in Equation (13) by alternately conducting the E and M steps.

• E step:

In the E step, we calculate the conditional expectation $\Upsilon_{ijk}$ of $Z_{ijk}$. The specific calculation formula can be given as follows.

(17)$$\Upsilon_{ijk}=E\left(Z_{ijk}\right)=\;\frac{\pi_{k}N\left(M_{ij}-u_{i}{}^{T}v_{j}-R_{ij}-C_{ij},0,\sigma_{k}^{2}\right)}{\sum_{k=1}^{n_{c}}\pi_{k}N\left(M_{ij}-u_{i}{}^{T}v_{j}-R_{ij}-C_{ij},0,\sigma_{k}^{2}\right)}$$

• M step:

In the M step, we need to optimize the following optimization model as
(18)$$\underset{U,V,R,C,\Pi ,\Sigma }{\min}\lambda_{1}{\left||R|\right|}_{2,1}+\lambda_{2}||C^{T}||{}_{2,1}\;-\sum_{i,j\in \Omega }\sum_{k=1}^{n_{c}}\Upsilon_{ijk}\left(log\pi_{k}-log\sqrt{2\pi }\sigma_{k}-\frac{1}{2\sigma_{k}^{2}}{\left(M_{ij}-u_{i}{}^{T}v_{j}-R_{ij}-C_{ij}\right)}^{2}\right)\;s.\;t.\;\;\sum_{k=1}^{n_{c}}\pi_{k}=1,\pi_{k}\geq 0$$

A natural idea is to use alternate iteration methods to solve MoG parameters Π,Σ and matrix variables U,V,R,C, i.e.,

**Update**Π,Σ:(19)$$\pi_{k}=B_{k}/\sum_{k=1}^{n_{c}}B_{k},\;\mathrm{where}\;B_{k}=\sum_{i,j\in \Omega }\Upsilon_{ijk}$$(20)$$\sigma_{k}^{2}=\frac{1}{B_{k}}\sum_{i,j\in \Omega }\Upsilon_{ijk}{\left(M_{ij}-u_{i}{}^{T}v_{j}-R_{ij}-C_{ij}\right)}^{2}$$

**Update**U,V,R,C:

In order to update U,V,R,C**,** we need to solve the following sub-problem:(21)$$\underset{U,V,R,C}{\min}\lambda_{1}{\left||R|\right|}_{2,1}+\lambda_{2}||C^{T}||{}_{2,1}+\underset{i,j\in \Omega }{\sum }\sum_{k=1}^{n_{c}}\Upsilon_{ijk}\frac{{\left(M_{ij}-u_{i}{}^{T}v_{j}-R_{ij}-C_{ij}\right)}^{2}}{2\sigma_{k}^{2}}$$

If we let
(22)$$W_{ij}=\{\begin{array}{c} \sqrt{\sum_{k=1}^{n_{c}}\Upsilon_{ijk}/\sigma_{k}^{2}},\left(i,j\right)\in \Omega ,\\ \;\;\;\;\;\;\;\;\;\;\;\;\;\;\;0\;\;\;\;\;\;\;\;\;,\;\;otherwise. \end{array}$$

Then Equation (21) can be reformulated as follows:(23)$$\underset{U,V,R,C}{\min}\lambda_{1}{\left||R|\right|}_{2,1}+\lambda_{2}||C^{T}||{}_{2,1}+\frac{1}{2}||W⊙{\left(M-UV^{T}-R-C\right)||}_{F}^{2}$$

Furthermore, we can optimize Equation (23) by solving the following three sub-problems.

1) Update U,V by solve the following sub-problem:(24)$$\underset{U,V}{\min}\frac{1}{2}||W⊙{\left(M-UV^{T}-R-C\right)||}_{F}^{2}$$

Obviously, as a typical weighted matrix decomposition problem, it could be solved by various methods, such as ALS [30], WLRA [31] and DN [32].

2) Update R by conducting the following iteration:(25)$$R_{t}=J_{\lambda_{1}\tau_{R}}\left(R_{t-1}-\tau_{R}W⊙W⊙\left(R_{t-1}+C+UV^{T}-M\right)\right)$$
where t represents the t-th iteration and $\tau_{R}$ is the step size of the proximal gradient.

3) Update C by conducting the following iteration:(26)$${\left(C^{T}\right)}_{t}=J_{\lambda_{2}\tau_{C}}\left({\left(C_{t-1}-\tau_{c}W⊙W⊙\left(R+C_{t-1}+UV^{T}-M\right)\right)}^{T}\right)$$
where t represents the t-th iteration and $\tau_{C}$ is the step size of the proximal gradient.

Based on the aforementioned analysis, we summarize the whole procedure in Algorithm 1.

**Algorithm 1.** Proposed Robust $\ell_{2,1}$-norm Regularized Matrix Decomposition (RRMD) Algorithm**Input:** The sampled EDM matrix M, the index set Ω, the parameters $\lambda_{1}$, $\lambda_{2}$ and threshold θ, the initial number of Gaussian components $n_{c}$.**Output:** The true underlying EDM matrix $\hat{D}$, the row structural anomaly R, and the column structural anomaly C.1. Randomly initialize $U_{0},V_{0},\Pi_{0},\Sigma_{0}$; initialize $R_{0},C_{0}$ to zero matrix;2. t=1;3. While not convergence do4.  (E-step)**:** update Υ according to Equation (17);5.  (M-step)**:** update Π,Σ according to Equation (19) and Equation (20);6.  (M-step): update U,V according to optimize Equation (24);7.  (M-step): update R according to Equation (25);8.  (M-step): update C according to Equation (26);9.  (Tuning $n_{c}$): Let $g_{i}$ and $g_{i}$ represent the number of i-th and j-th Gaussian component respectively. if $\left|\sigma_{i}{}^{2}-\sigma_{j}{}^{2}\right|/\left(\sigma_{i}{}^{2}+\sigma_{j}{}^{2}\right)<\theta$, then let $\pi_{i}=\pi_{i}+\pi_{j}$, $\sigma_{i}{}^{2}=\left(g_{i}\sigma_{i}{}^{2}+g_{j}\sigma_{j}{}^{2})/(g_{i}+g_{j}\right)$, K=K−1. Lastly, remove $\pi_{j}$ and $\sigma_{j}^{2}$ from Π,Σ, respectively.10.  t=t+1;11. End while 12. $R,C\;\mathrm{and}\;\hat{D}=UV^{T}$.

### 4.2. Anomaly-Aware Node Localization

Based on the proposed RRMD model, all pair-wise range measurements between nodes can be easily obtained. However, it is only the first step of this range-based localization method. The second step of this method is to calculate the location information of the unknown nodes based on the complete distance information and the actual coordinates of anchor nodes, and which can be implemented by using the classical MDS method [6,8,21].

MDS is a typical low-dimensional embedding method, which is originally proposed to solve the curse of dimensionality in the field of machine learning. The MDS method ensures that the distance between samples in the original high-dimensional space can be preserved in low-dimensional space. The input of this method can be high-dimensional features of samples (the pair-wise distance information can be easily obtained from high-dimensional features) or the distance information between samples, while the output is the low-dimensional features of samples in the specified d-dimensional space. Therefore, in the sensor networks localization application, we can employ the MDS method to calculate the relative coordinates of each node in the d-dimensional space. Then, based on the actual/absolute coordinates of at least d+1 anchor nodes and the relative coordinates obtained, the coordinate transformation matrix between the actual coordinates and the relative coordinates can be calculated (with the details provided in Theorem 2). Finally, the relative coordinates can be mapped to the actual ones by using the coordinate transformation matrix.

Next, we introduce the specific steps to implement the above MDS method. Firstly, we let the relative coordinates and actual coordinates of n nodes be represented as T=[t1,t2,…,tn]∈ℝd×n, A=[a1,a2,…,an]∈ℝd×n, respectively, and assume that nodes 1,2,…na(na≥d+1) are anchor nodes. Secondly, the classic MDS method is employed to calculate the relative coordinates of all nodes. Thirdly, with the actual coordinates of anchor nodes, we have coordinate transformation matrix:(27)Q=[a2−a1,a3−a1,…,ana−a1][t2−t1,t3−t1,…,tna−t1]

Finally, the coordinate of all unknown nodes can be obtained by
(28)$$(a_{i}=Q\cdot \left(t_{i}-t_{1}\right)+a_{1},i=n_{a}+1,n_{a}+2,\ldots n$$

Based on the aforementioned analysis, the proposed Anomaly-aware Node Localization (ANLoC) algorithm can be summarized as in Algorithm 2.

**Algorithm 2.** Anomaly-aware Node Localization (ANLoC) Algorithm**Input:** The sampled EDM matrix M, the index set Ω, the parameters λ1, λ2 and threshold θ, the initial number of Gaussian components nc. The coordinates of anchor nodes {a1,a2,…,ana}(na≥d+1), where na denotes the number of anchor nodes.**Output:** Coordinates of all unknown nodes  {ai|i=na+1,na+2,…n}.1. Calculate the true underlying EDM matrix D^ by using Algorithm 1;2. Double centering the matrix D^:  S=−0.5×JD^J, where J=I−1·1T/n and I is identity matrix;3. Perform SVD decomposition on matrix S:[H,Λ,Κ]=svd(S);4. Calculate relative coordinates: 
T=[t1,t2,…,tn]=Λd·HdT
   where ti∈ℝd×1, Λd= Λ(1:d,1:d), Hd=H(:,1:d);5. Calculate the coordinate transformation matrix Q:  Q=[a2−a1,a3−a1,…,ana−a1]/[t2−t1,t3−t1,…,tna−t1];6. Calculate and output coordinates of all unknown nodes: 
ai=Q·(ti−t1)+a1,i=na+1,na+2,…n.


**Theorem** **2:** *For sensor nodes localization in*d*-dimensional space, given the absolute positions*$\left\{a_{1},\;a_{2},\cdots ,a_{n_{a}}\right\}$*of the*$n_{a}$*anchor nodes and the relative coordinates*$\left\{t_{1},\;t_{2},\cdots ,t_{n}\right\}$*of all the*$n\;\left(n\gg n_{a}\right)$*sensor nodes, if*$n_{a}\geq d+1$*, then the relative coordinates*$\left\{t_{n_{a}+1},\;t_{n_{a}+2},\cdots ,t_{n}\right\}$*can be transformed to the corresponding absolute positions*$\left\{a_{n_{a}+1},\;a_{n_{a}+2},\cdots ,a_{n}\right\}$.

**Proof:** According to reference [13], the absolute positions ai (i≥na+1) can be computed according to
(29)ai=Q·(ti−t1)+a1,i=na+1,na+2,⋯,n
where Q∈ℝd×d is the unknown coordinate-transform matrix, which should be determined by the following matrix equation:(30)Q·[t2−t1,t3−t1,⋯,tna−t1]=[a2−a1,a3−a1,⋯,ana−a1]Without loss of generality, let T^=[t2−t1,t3−t1,⋯,tna−t1] and A^=[a2−a1,a3−a1,⋯,ana−a1], then we can see that T^∈ℝd×(na−1) and A^∈ℝd×(na−1). Therefore, the above matrix Equation can be regarded as the following equivalent equations:(31)∑l=1dQil·T^lj=A^ij,i=1,2,⋯,d, and j=1,2,⋯,na−1Obviously, if na≤d, then the number of unknown entries (i.e., d2) is greater than the number of Equations (i.e., d×(na−1)), so we cannot obtain the determined coordinate-transform matrix Q. On the contrary, if na≥d+1, we only need to arbitrarily select d+1 anchor nodes, then we can obtain the unique solution of Q. □

## 5. Performance Evaluation

In this section, we first describe the experimental setting and evaluation metrics, and then report some extensive experimental results under the different scenarios.

### 5.1. Experimental Setting

In order to investigate the performance of our proposed ANLoC, some simulation experiments were conducted. All these experiments were designed based on MATLAB 2017a and run on PC with Intel i5-8400 CPU and 16G RAM. Firstly, we randomly disposed 100 sensor nodes in a square area with 100 × 100 unit (where the unit can be determined according to actual communication condition, such as meter, decimeter, foot, and inch, etc.), and 6 of them were anchor nodes while others were unknown nodes. Let X∈ℝ2×100 be the actual coordinate matrix of all nodes and D∈ℝ100×100 denote the ground truth EDM matrix between nodes. Secondly, we artificially added some error matrices into matrix D to simulate the complex noises and the structural anomaly, and thus obtained a corrupted matrix Derror. Specifically, the complex noise was set to the mixture of the following components: (1) Gaussian noise with mean 0 and variance 100; (2) Gaussian noise with mean 0 and variance 50; and (3) sparse noise with a pollution ratio 1% randomly generated within the range of [0,10000]. Moreover, the structural anomalies were set to be randomly generated within the range of [0,500] and with a row-wise pollution ratio 3% and a column-wise pollution ratio 3%. Thirdly, sampled matrix M could be obtained by randomly sampling some elements from Derror. Finally, the true underlying EDM matrix can be reconstructed from M via Algorithm 1, and then the actual coordinates of all nodes could be calculated using Algorithm 2. Moreover, we summarized all the simulation parameters in Table 1.

The proposed ANLoC method involves three hyper-parameters $\lambda_{1},\;\;\lambda_{2}$ and $n_{c}$. It is important to adjust these parameters to proper values. In order to determine the optimal value of these parameters, we employed a 10-fold cross validation procedure and conduct the grid search method. In particular, firstly, let $n_{c}=5$, we searched for the optimal solution of $\lambda_{1}$ and $\lambda_{2}$ within the range of {0.001, 0.01, 0.1, 1, 10,100,1000} by minimizing the matrix recovery error of EDM. Then, based on the selected optimal parameter $\lambda_{1}$ and $\lambda_{2}$, $n_{c}$ was determined within the range of {1, 2, 3, 4, 5, 6, 7, 8, 9, 10}. Last but not least, we compared our method with three different competing methods, including NLIRM method [6], SVT-based method [20], and OptSpace-based method [33]. All the involved parameters in these competing methods were optimized by using the same nested 10-fold cross-validation procedure as in our ANLoC method.

### 5.2. Evaluation Metrics

We selected the following four evaluation metrics to evaluate the node localization performance:
EDM recovery error:(32)$$e_{r}=||\hat{D}-D|{|}_{F}/{\left||D|\right|}_{F}$$
where $\hat{D}$ denotes the reconstructed matrix and D is the ground truth matrix.Localization error:(33)$$e_{l}=||\hat{X}-{X||}_{F}/n$$
where $\hat{X}$ denotes the coordinates calculated by Algorithm 2.Anomaly recognition accuracy:(34)$$p_{ave}=\left(p_{row}+p_{col}\right)/2$$
where $p_{row}$ and $p_{col}$ represent the row anomaly recognition accuracy and column anomaly recognition accuracy, respectively. Specifically, the $p_{row}$ is calculated by
(35)$$p_{row}=2\times \frac{r_{pre}\cdot r_{rec}}{r_{pre}+r_{rec}},r_{pre}=\frac{r_{tru}}{r_{all}},r_{rec}=\frac{r_{tru}}{r_{act}}$$
where $r_{all}$ represents the number of abnormal rows identified by the method proposed in this paper, $r_{tru}$ indicates the number of correctly recognized rows in $r_{all}$ and $r_{act}$ represents the number of actual abnormal rows; the $p_{row}$ is calculated by
(36)$$p_{col}=2\times \frac{c_{pre}\cdot r_{rec}}{c_{pre}+r_{rec}},c_{pre}=\frac{c_{tru}}{c_{all}},c_{rec}=\frac{c_{tru}}{c_{act}}$$
where $c_{all}$ represents the number of abnormal columns identified by the method proposed in this paper, $c_{tru}$ indicates the number of correctly recognized columns in $c_{all},$ and $c_{act}$ represents the number of actual abnormal columns. Cumulative distribution of localization errors [21]: (37)$$e_{CD}=P\left(\Delta l_{i}\leq \sigma \right)$$
where $\Delta l_{i}=\sqrt{{\left({\hat{x}}_{i}-x_{i}\right)}^{2}+{\left({\hat{y}}_{i}-y_{i}\right)}^{2}}$ defines the localization error of i-th node, $\left(x_{i},y_{i}\right)$ and $\left({\hat{x}}_{i},{\hat{y}}_{i}\right)$ denote the actual coordinates and the estimated coordinates, respectively.

### 5.3. Experimental Results

In order to investigate the performance of proposed method in complex environments, we conducted the following four experiments under the different application scenarios.

• Scenario 1: Localization without the noise and anomaly

This experiment assumed that all observed range measurements were accurate. Figure 3 shows that the variation of the EDM recovery error and average localization error of WSNs nodes when the EDM is sampled at different proportions. The horizontal axis of the two subgraphs represents the sampling ratio, while the vertical axis in Figure 3a represents the EDM recovery error, and the vertical axis in Figure 3b represents the localization error of WSNs nodes. Comparing Figure 3a with Figure 3b, it is obvious to find that the EDM recovery error and localization error are consistent with the trend of the sampling rate. Under this scenario, when sampling rate reaches 0.3, the EDM recovery error and the localization error of these four methods reach a very low level. This shows that the existing methods and the method proposed in this paper can achieve good positioning accuracy under the ideal condition.

• Scenario 2: Localization with only complex noise

This experiment aims to examine the performance of four methods with only complex noise. Therefore, we only add complex noise into EDM. After sampling the EDM in different proportions, the experimental results are shown in Figure 4. Under this scenario, the SVT-based method and the OptSpace-based method have large EDM recovery error and localization error even if the sampling rate is high, which indicates that the two methods do not handle the complex noise well. When the sampling ratio reaches 0.3, the EDM recovery error of NLIRM could be stabilized at around 0.01, and the localization error of nodes could be less than 0.03. However, the EDM recovery error and localization error of ANLoC could reach 0.001 and 0.01, respectively. Comparing with Figure 3, we find that both the SVT-based method and the OptSpace-based method could not deal with complex noise at all. The NLIRM has a certain ability to resist noise, but it is slightly worse than our ANLoC. Therefore, our ANLoC could perform best under complex noise than others.

• Scenario 3: Localization with only anomaly

The purpose of this experiment is to examine the performance of detecting anomalous nodes in the environment without any noise. In this case, only anomaly is added into EDM. As shown in Figure 5, compared with Figure 3, the EDM recovery error and the localization error of the four methods are all increased when the EDM is destroyed by the anomaly. However, compared with other three methods, ANLoC could have better performance at a lower sampling rate (0.2). In addition, the ANLoC could also detect abnormal nodes. Specifically, Figure 5c shows that when the sampling rate reaches 0.2, the recognition accuracy of abnormal nodes can reach 100%. In comparison with Figure 3, we find that the performance of all methods has been degraded in varying degrees, but our ANLoC still has the best performance. Therefore, when some sensor nodes have abnormities, the ANLoC has good performance for precisely locating unknown nodes and detecting abnormal nodes.

• Scenario 4: Localization with both complex noise and anomaly

This experiment is designed to evaluate the performance of the ANLoC in a complex environment where complex noise and anomaly coexist. The complex noise and the anomaly are both added to the EDM to simulate the impact of the complex real application scenarios. Experimental results are shown in Figure 6. Obviously, compared with the other three methods, our ANLoC method proposed in this paper not only has the lowest EDM recovery error and localization error, but also has high recognition accuracy of abnormal nodes when the sampling rate reaches 0.3. Comparing with Figure 3, we can find that the performance of the SVT-based method and the OptSpace-based method are much worser in complex environment. The NLIRM has less performance degradation, but its EDM recovery accuracy and localization accuracy are still not as high as our ANLoC. In short, our proposed ANLoC has good robustness to complex application scenarios and could detect faulty nodes well. In addition, Figure 7a shows the cumulative distribution of the localization error for the four methods when the sampling rate is 0.5 under this scenario. The probability of the localization error of the ANLoC being less than 0.5 is 98% and the probability of being less than 1 is 100% while the other three methods are all below 30%. In addition, Figure 7b shows the intuitionistic localization results of all nodes in this case when the sampling rate is 0.5.

### 5.4. Effects of the Proposed Strategies

To analyze the effects of our proposed strategies, experiments are designed to compare the proposed method with three partial deleted methods. Actually, our RRMD can be treated as LRMD + MoG + $\ell_{2,1}$-norm. Then, the remaining three types of partial deleted methods are as follows. (1) Pure LRMD: do not consider any noise or anomaly; (2) LRMD + MoG: only consider complex noise; (3) LRMD + $\ell_{2,1}$-norm: only consider the anomaly. What’s more, these experiments are set under the scenario with both complex noise and anomaly. Finally, all experimental results are shown in Figure 8a–c. Obviously, our ANLoC considers both complex noise and anomaly, which leads to its low EDM recovery error and localization error. Moreover, when the sampling rate reaches 0.3, it could accurately detect abnormal nodes. Due to too ideal assumptions, pure LRMD is the worst performer, which means that it cannot be applied in actual situation. Although LRMD + MoG could handle the error of EDM well, it does not have the ability to detect abnormal nodes. Specially, LRMD + $\ell_{2,1}$-norm could hardly detect abnormal nodes in this case. Naturally, we speculate the noise that is not unprocessed may have a negative effect on the performance for detecting abnormal nodes in this method. Therefore, another experiment is designed with LRMD + $\ell_{2,1}$-norm when the EDM is only destroyed by anomaly, and the result of anomaly recognition is drawn in Figure 8d. The pleasure is that this supplementary experiment validates our conjecture. In conclusion, these experiments demonstrate the effectiveness of the MoG and $\ell_{2,1}$-norm strategies.

### 5.5. Localization for Large-Scale Scenario

In a practical application scenario, there is a case that when the size of the localization scenario is much larger than ranging length of the sensor nodes and only a few range measurements between inter-nodes could be obtained. As a result of that, the sampled EDM based on range measurements will be very sparse, which leads to poor performance of our method. To solve this problem, a large-scale localization method is proposed in this paper. For the convenience of discussion, we simplify the application scenario as shown in Figure 9. Suppose there is a 180×100 unit area divided into three parts I, II, and III from left to right, and among them, there are 80, 20, 80 nodes, respectively. Of the 180 nodes, there are 6 anchor nodes and 6 relay nodes. Then, regions I and II are combined into a localization sub-region M while regions II and III are combined into a localization sub-region N. The large-scale process is as follows. (1) Calculate the actual coordinates of all nodes within sub-region M by ANLoC. (2) Based on step one, the actual coordinates of relay nodes are gained. Taking them as anchor nodes, we employ ANLoC again to get the coordinates of the remaining unknown nodes in region N.

Further, in order to verify the feasibility of the above-mentioned large-scale scenario localization method, we designed an experiment with complex noise and anomaly. Moreover, the EDM sampling rates of region M and N are both set 0.5. As Figure 10 shows, the experimental results confirm that our large-scale localization method can be well applied to large-scale scenarios.

## 6. Conclusions

In this paper, we aim to estimate node position of WSNs in complex environments. Considering that the coexistence of potential node anomaly and complex noise in practical applications, we propose a novel Anomaly-aware Node Localization (ANLoC) method to address this task. Specifically, the proposed ANLoC method is divided into two steps. First, a Robust $\ell_{2,1}$-norm Regularized Matrix Decomposition (RRMD) model is designed to simultaneously detect node anomaly and impute the missing range measurements. Second, based on the imputed EDM matrix, all unknown nodes can be easily estimated by using the classic MDS method. The extensive experiments demonstrated our proposed ANLoC method consistently outperforms three state-of-the-art localization methods in terms of EDM recovery error, localization error, and anomaly recognition accuracy. Additionally, our proposed ANLoC method also can extend to a large-scale localization scenario. Future work will focus on extending our ANLoC approach to an incremental version, making it suitable for handling dynamic positioning of mobile sensor nodes.

## Figures and Tables

**Figure 1 sensors-19-01912-f001:**
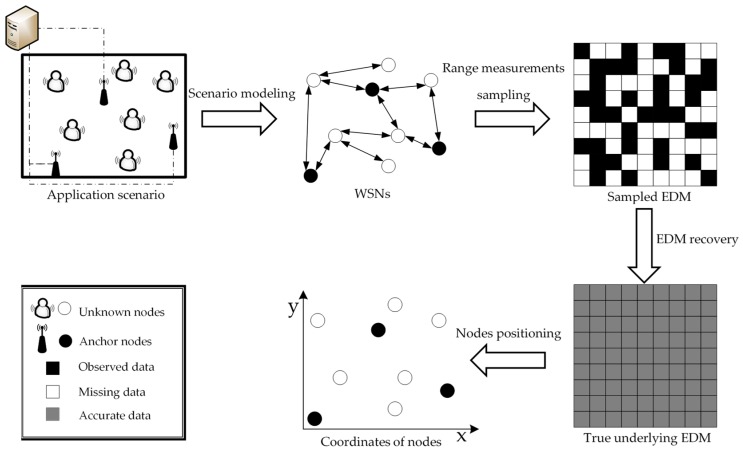
Pipeline of node localization for Wireless Sensor Networks (WSNs) in complex environments.

**Figure 2 sensors-19-01912-f002:**
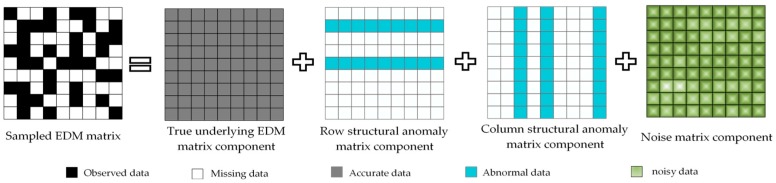
Illustration of the sampled Euclidean Distance Matrix (EDM) matrix.

**Figure 3 sensors-19-01912-f003:**
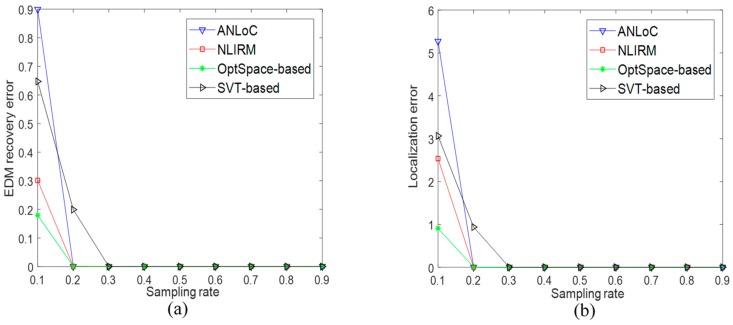
Performance comparison under scenario 1 without the noise and anomaly. (**a**) EDM recovery error and (**b**) localization error.

**Figure 4 sensors-19-01912-f004:**
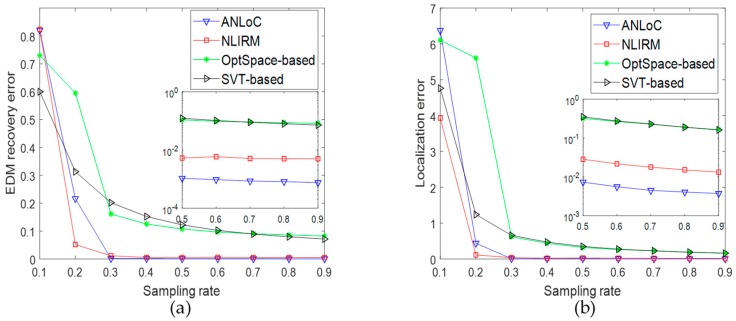
Performance comparison under scenario 2 with only complex noise. (**a**) EDM recovery error and (**b**) localization error.

**Figure 5 sensors-19-01912-f005:**
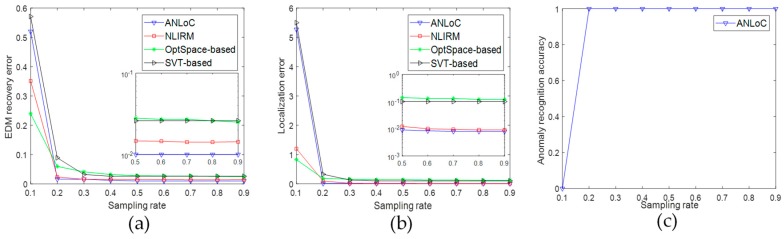
Performance comparison under scenario 3 with only anomaly. (**a**) EDM recovery error; (**b**) localization error; and (**c**) anomaly recognition accuracy.

**Figure 6 sensors-19-01912-f006:**
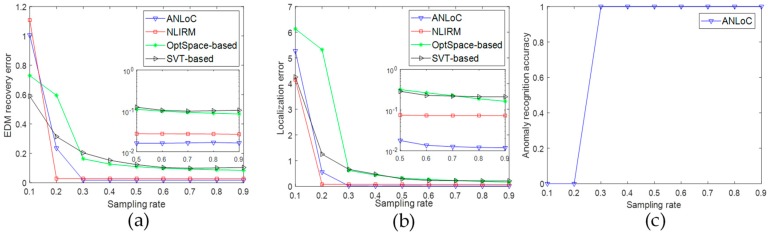
Performance comparison under scenario 4 with both complex noise and anomaly. (**a**) EDM recovery error; (**b**) localization error; and (**c**) anomaly recognition accuracy.

**Figure 7 sensors-19-01912-f007:**
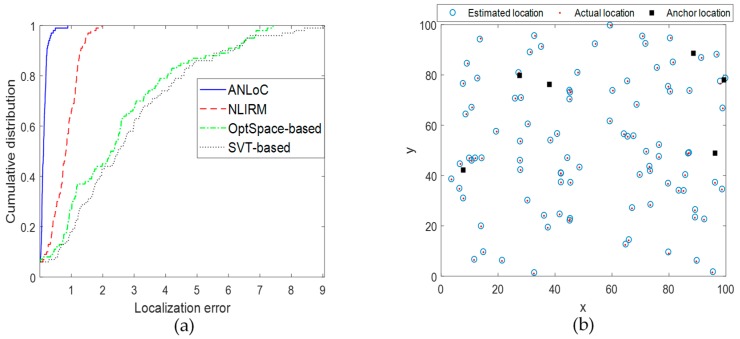
Illustration of localization performance under scenario 4 with both complex noise and anomaly. (**a**) Cumulative distribution of localization error and (**b**) positioning result display.

**Figure 8 sensors-19-01912-f008:**
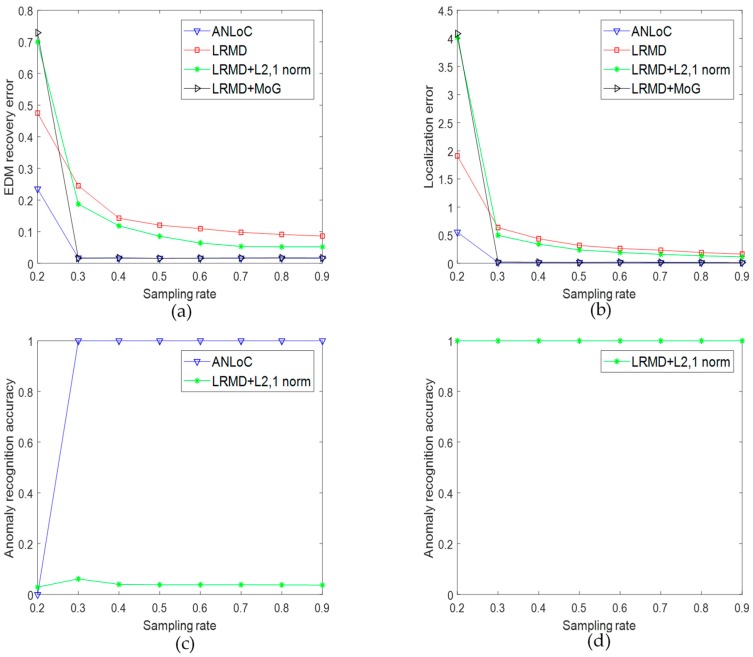
Effects illustration of the proposed strategies. (**a**) EDM recovery error under the scenario 4 with both complex noise and anomaly; (**b**) localization error under the scenario 4 with complex noise and anomaly; (**c**) anomaly recognition accuracy under condition with complex noise and anomaly; and (**d**) anomaly recognition accuracy under the scenario 3 with only anomaly.

**Figure 9 sensors-19-01912-f009:**
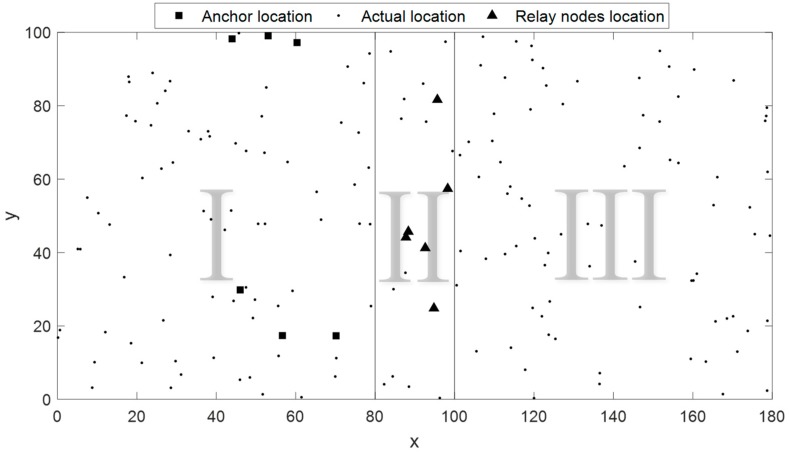
Illustration of large-scale scenario used in this study.

**Figure 10 sensors-19-01912-f010:**
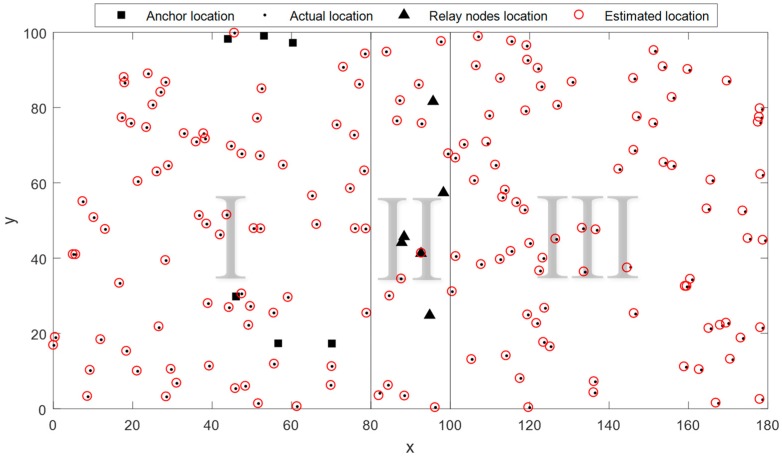
Localization results under large-scale scene scenario with complex noise and anomaly.

**Table 1 sensors-19-01912-t001:** Simulation parameters.

Name	Value	Name	Value
Size of experimental scenario	100 × 100	Gaussian noise 1	Mean 0, variance 100
Number of sensor nodes	100	Gaussian noise 2	Mean 0, variance 50
Number of anchors	6	Range of sparse noise	[0,10000]
Range of row anomaly	[0,500]	Range of column anomaly	[0,500]

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
