# Peer review of "ANLoC: An Anomaly-Aware Node Localization Algorithm for WSNs in Complex Environments"

_sensors, 2019, doi:10.3390/s19081912_

Round 1

Reviewer 1 Report

The article presents the problem of node localization  for Wireless Sensor Networks in Complex Environments. This problem is specially challenging when accurate localization is needed in environments affected by  wider noise distributions.

This problem is relevant for many applications that require node localization or  location verification. The article is general is well written and it clearly states the details of the problems and novelty of the proposed solutions.  However, I can offer a few comments:

1) Figure 1 should be explained a little more to explain the meaning of the Sampled EDM and true underlying EDM figure

2) I understand that the meaning of EDM is Euclidean Distance Matrix, this meaning should be stated in the article.

3) The article shows several results in the experimental section which in some cases are quite close the baseline approaches. It should be interesting to see whether there exists a significant  statistical difference among the results

Author Response

We would like to thank  reviewer for his/her helpful comments and suggestions. In the revised submission, we have addressed all these comments. In the uploaded file, we provided point-by-point responses to the comments of the reviewer and describe how they are handled in the revised submission. Also, the revisions in the revised submission are marked with red font.

Reviewer 2 Report

The paper presents an anomaly-aware localization scheme that imputes missing range measurements and detects anomaly in complex environments. After formulating the range measurement imputation problem, an algorithm is designed for solving the derived model by the formulation. The performance of the proposed scheme is evaluated and compared via the Matlab simulation.

The novelty and originality of the paper is fairly good. The organization and presentation of the paper are relatively well addressed. For readers to catch the simulation parameters easily, however, it is suggested to summarize the simulation parameters in a table in Section 5.1. In addition, it is much better if some possible future works are introduced in the Conclusions section.

Author Response

(The authors gave the same response as above.)

Reviewer 3 Report

The paper proposes an anomaly-aware node localization algorithm for WSNs that takes into account complex noise distribution and the presence of node anomaly. The complex noise is modeled using the Mixture of Gaussian (MoG) distribution, and using the Robust l2,1-norm Regularized Matrix Decomposition (RRMD) model, a fully populated Euclidean Distance Matrix (EDM) is derived which can be used with Multi-Dimensional Scaling to estimate the locations of the unknown nodes in the WSN. The proposed scheme is interesting and has some novelty. Experimental results are obtained and the schemes is shown to perform better compared to 3 other localization methods. Detailed comments as follows:

Lines 175 - 177: It is stated that bold upper case is used to denote matrices. So the use of Xi,j as scalar seems confusing.

Line 227: It is stated that D denotes the true EDM. However, in practical situations, the positions of most nodes are unknown except for the limited number of anchor or reference nodes. Indeed in eqn. (13), D is required in the RRMD model. How is this possible? Doesn't this undermine the entire RRMD model?

Algorithm 1 is used to derive the complete EDM matrix D. But this matrix D cannot the same as the one defined in line 227 above in which the latter denotes the true EDM. This is confusing. It would seem that Algorithm 1 is deriving the estimated D instead of the true D. IF so, this should be clearly stated, and notation to differentiate them accordingly.

In all the figures that show the localization errors, what is the units? Meters? Please specify.

Line 443: It is stated that "all observed range measurements are accurate". This assumption is flawed and not practicable in real life situations. If so, then the experimental results are idealistic and of limited relevance. Please comment.

Section 5.5: The experimental scenario for the large-sale scene scenario excludes noise and anomalies. This is unrealistic and thus presents an idealistic scenario. It would have been more meaningful to include noise, at the very least, and anomalies too to provide more meaningful insights to validate applicability to large-scale scenarios.

Author Response

(The authors gave the same response as above.)

Round 2

Reviewer 3 Report

The authors have adequately addressed the reviewers' comments. Changes and additions have been made to improve the clarity and scope of the paper.